# Vinorelbine Augments Radiotherapy in Hepatocellular Carcinoma

**DOI:** 10.3390/cancers12040872

**Published:** 2020-04-03

**Authors:** Kheng Wei Yeoh, Aldo Prawira, Muhammad Zafrie Bin Saad, Kok Ming Lee, Eric Ming Hon Lee, Gee Keng Low, Mohamed Hakim Bin Mohd Nasir, Jun Hao Phua, Wendy Wan Li Chow, Iris Jiu Hia Lim, Yusnita Binte Omar, Rebecca Zhi Wen Ho, Thi Bich Uyen Le, Thanh Chung Vu, Khee Chee Soo, Hung Huynh

**Affiliations:** 1Department of Radiation Oncology, Singapore General Hospital, Outram Road, Singapore 169608, Singapore; yeoh.kheng.wei@singhealth.com.sg (K.W.Y.); muhd.zafrie.saad@nccs.com.sg (M.Z.B.S.); lee.kok.ming@nccs.com.sg (K.M.L.); eric.lee.m.h@nccs.com.sg (E.M.H.L.); low.gee.keng@nccs.com.sg (G.K.L.); mohd.hakim.mohd.nasir@nccs.com.sg (M.H.B.M.N.); phua.jun.hao@nccs.com.sg (J.H.P.); trdcwl@nccs.com.sg (W.W.L.C.); irisjiuhialim@yahoo.com.sg (I.J.H.L.); trdybo@nccs.com.sg (Y.B.O.); 2Laboratory of Molecular Endocrinology, Division of Molecular and Cellular Research, National Cancer Centre, Singapore 169610, Singapore; aldo.prawira@nccs.com.sg (A.P.); rebecca.ho.z.w@nccs.com.sg (R.Z.W.H.); uyenbright3011@gmail.com (T.B.U.L.); vu.thanh.chung@nccs.com.sg (T.C.V.); 3Tan Chin Tuan Laboratory of Optical Imaging and Photodynamic & Proton Therapy, Division of Medical Sciences, National Cancer Centre, Singapore 169610, Singapore; soo.khee.chee@singhealth.com.sg

**Keywords:** hepatocellular carcinoma, radiotherapy, radio sensitization, vinorelbine, DNA repair, cost-effectiveness

## Abstract

There is a need to improve the effectiveness of radiotherapy (RT) in hepatocellular carcinoma (HCC). Therefore, the purpose of this study was to explore the efficacy and toxicity of the anti-microtubule agent Vinorelbine as a radiosensitizer in HCC. The radio sensitivity of 16 HCC patient-derived xenograft (PDX) models was determined by quantifying the survival fraction following irradiation in vitro, and Vinorelbine radio sensitization was determined by clonogenic assay. Ectopic HCC xenografts were treated with a single dose of 8 Gy irradiation and twice-weekly 3 mg/kg Vinorelbine. Tumor growth and changes in the proteins involved in DNA repair, angiogenesis, tumor cell proliferation, and survival were assessed, and the 3/16 (18.75%), 7/16 (43.75%), and 6/16 (37.5%) HCC lines were classified as sensitive, moderately sensitive, and resistant, respectively. The combination of RT and Vinorelbine significantly inhibited tumor growth, DNA repair proteins, angiogenesis, and cell proliferation, and promoted more apoptosis compared with RT or Vinorelbine treatment alone. Vinorelbine improved HCC tumor response to standard irradiation with no increase in toxicity. HCC is prevalent in less developed parts of the world and is mostly unresectable on presentation. Vinorelbine and conventional radiotherapy are cost-effective, well-established modalities of cancer treatment that are readily available. Therefore, this strategy can potentially address an unmet clinical need, warranting further investigation in early-phase clinical trials.

## 1. Introduction

Liver cancer was the sixth most prevalent cancer in 2018 and the fourth leading cause of cancer-related mortality globally, with a higher prevalence in less-developed parts of the world. The vast majority of cases are hepatocellular carcinoma (HCC), which carries a 5-year overall survival (OS) rate of around 5% [1]. The only potential curative therapy for HCC is surgical resection and liver transplantation; however, only 10–20% of cases are deemed operable at presentation [2,3,4]. For unresectable HCC, prognosis with current loco-regional, systemic, and molecularly targeted approaches remains poor [5,6,7,8,9,10]. The multikinase inhibitor Sorafenib has been established as a standard therapy for patients with advanced HCC, having been shown to improve median survival by only around 3 months [11,12]. Other multikinase inhibitors, such as Regorafenib, Carbozantinib, and Lenvatinib, have also been shown to benefit this group of patients [13,14,15]. Furthermore, immunotherapy has shown encouraging early results for HCC patients, with response rates of around 20%, prompting cautious optimism with ongoing clinical trials [16,17]. However, a randomized phase III study evaluating Nivolumab versus Sorafenib as a first-line treatment in patients with unresectable HCC (NCT02576509) has failed to achieve its primary endpoint of OS. There is, therefore, an unmet clinical need for this group of patients.

Radiotherapy (RT) is a well-established and cost-effective modality of cancer treatment, with over 50% of all cancers requiring RT and accounting for only about 5% of treatment costs [18,19]. However, its utility for managing HCCs has been limited, mostly for palliation when other treatment options have been exhausted. In part, this has been due to its toxicity, where radiation-induced liver disease is observed at doses exceeding 30 Gy [20], prompting for more conformal techniques [5].

The advantages of RT, in combination with systemic therapy, have long been established in several cancer types, and RT is the standard of care for many advanced solid cancers. However, data for HCC with this strategy remain sparse. Of particular relevance, the Radiation Oncology Therapy Group (RTOG) 1112 randomized clinical trial is one such approach currently investigating the combination of RT with Sorafenib in HCC with the aim to improve the outcome of patients with this condition (NRG-RTOG 1112).

In this study, we investigated the effects of combining RT with Vinorelbine for HCC in vitro and in vivo to derive preclinical evidence for this approach. Vinorelbine, a semisynthetic vinca alkaloid that inhibits microtubule polymerization, has been shown to have radio-enhancer activity even at low doses [21,22]. Vinorelbine is also a cost-effective agent routinely used in clinical protocols for cancer patients being treated with concurrent chemoradiotherapy [23,24]. It is thus a potentially attractive strategy to utilize Vinorelbine and RT, at sub-toxic levels, for inoperable HCC patients. Moreover, we sought to better understand the molecular mechanisms underlying the antitumor effect of RT and its co-administration with Vinorelbine. Here, we report the effects of RT alone and an RT/Vinorelbine combination on tumor growth, tumor angiogenesis, DNA damage signaling pathways, cell proliferation, and apoptosis in human HCC patient-derived xenograft (PDX) mice models [25]. The findings indicate that the combination of RT and Vinorelbine is well tolerated and improves tumor response compared with RT alone, without increased toxicity, hence warranting further investigation in clinic.

## 2. Results

### 2.1. Screening of Organoid Cultures In Vitro and PDX Models in Mice for Sensitivity to Irradiation

The sensitivity of 16 HCC lines to RT was initially determined by subjecting organoid cultures to 8 Gy irradiation, and the 3/16 (18.75%), 7/16 (43.75%), and 6/16 (37.5%) HCC lines were classified as sensitive, moderately sensitive, and resistant to RT, respectively (Figure 1A). Furthermore, RT caused an increase in the percentage of cells in the G2M phase and a concomitant reduction in the percentage of cells in the G1 phase (Appendix A). We next investigated the potential of Vinorelbine as a radiosensitizer in vitro. Organoid cultures were irradiated at 6 Gy, which was shown to give a maximal inhibition in vitro in the presence or absence of 0.05 µM Vinorelbine. HCC17-0211, HCC13-0212, HCC01-0909, HCC13-0109, and HCC09-0913 cultures, which have varying degrees of sensitivity to RT, showed fewer colonies when Vinorelbine was present (Figure 1B).

Irradiation of HCC19-0913 tumor-bearing mice with 2, 8, and 20 Gy led to approximately 9% ± 7%, 84% ± 12%, and 83% ± 4% reductions in tumor burden relative to the non-irradiated control, respectively (Figure 1C). A single dose of 8 Gy significantly inhibited tumor growth without affecting the growth of the internal controls (left flanks) (Figure 2B; *p* = 5.04 × 10^−5^). At 20 Gy, the growth of the internal controls was partially inhibited compared with non-irradiated tumors (Figure 2C; *p* = 0.0128). A dose of 2 Gy was insufficient in eliciting long-term inhibition on tumor growth (Figure 2A; *p* = 0.0869), and no significant growth difference was observed when comparing between 8 Gy- and 20 Gy-treated groups (*p* = 0.6938). A dose of 8 Gy was deemed efficacious with minimal RT-associated toxicities in vivo, and, therefore, was chosen for subsequent RT/Vinorelbine combination studies. Similarly, 8 Gy also showed inhibition of tumor growth in the HCC17-0211, HCC13-0109, and HCC01-0909 lines compared with the control group (Figure 1D–F).

### 2.2. Effects of Localized Irradiation on Tumor Growth, Angiogenesis, DNA Damage, Apoptosis, and Cell Proliferation

Immunostaining of HCC19-0913 tumors from control and RT-treated mice (Figure 2A) revealed that tumors irradiated with 8 Gy showed a 2–2.8-fold higher proportion of p-Histone H3 Ser10-positive cells (Figure 2B) compared with left flank tumors, indicating that RT arrests cells at mitosis [26]. Furthermore, there was a 29–30-fold increase in cleaved PARP-positive cells (Figure 2C) and a 140–150-fold increase in p-Histone H2AX Ser139 (γH2AX)-positive cells (Figure 2D). These results suggest that RT caused DNA damage and induced apoptosis in HCC cells. However, there was no significant difference in the number of cells staining positively for p-Histone H3 Ser10 (Figure 2B), cleaved PARP (Figure 2C), or γH2AX (Figure 2D) between the left flank control and the 2 Gy-irradiated tumors. Morphologically, cells and nuclei in HCC19-0913 tumors irradiated with 8 Gy were slightly larger compared with the controls.

To determine if RT alters angiogenesis, tumor sections were stained with CD31 antibody. Blood vessel density in the irradiated tumors significantly increased compared to that of the non-irradiated tumors (Figure 2A,E; *p* = 7.6 × 10^−5^). Structurally, the blood vessels in the irradiated tumors were smaller than those in the non-irradiated tumors (Figure 2A). HypoxyProbe staining was negative across a large section of the irradiated-tumors, indicating that the regions were well-oxygenated (Figure 2A). This indicates that RT stimulates new vessel formation and the recovery of the vasculature.

Western blot analysis revealed that the levels of p-ATR and p-ATM in 8 Gy-irradiated tumors moderately decreased, and the levels of p-Chk2 and γH2AX increased. The levels of cleaved caspase 3, cyclin B1, and p27 increased; however, p-Cdc2 levels were reduced, suggesting that RT induced apoptosis and cell cycle arrest. The levels of p-Erk1/2, p-Akt, p-mTOR, and the downstream targets of mTOR were not significantly affected by RT (Appendix A). A similar effect was not observed in 2 Gy-irradiated tumors, indicating that 2 Gy was insufficient to change the expression of the proteins involved in DNA damage, cell death, and cell proliferation. Similar data were obtained when HCC13-0212 tumors were analyzed (Appendix A).

### 2.3. Vinorelbine Potentiates Antitumor Activity of RT in HCC Models

Next, we examined whether the antitumor activity of RT is augmented by Vinorelbine. Mice were treated with 8 Gy irradiation, 3 mg/kg Vinorelbine, or a combination of both. A metronomic dose of 3 mg/kg Vinorelbine showed modest inhibition (*p* < 0.01), but RT potently inhibited the growth of HCC19-0913 tumors (Figure 3A; *p* < 0.001). However, compared with monotherapy, the combination of Vinorelbine/8 Gy irradiation inhibited tumor growth more completely (Figure 3A; *p* < 0.001), indicating that Vinorelbine acts in synergy with RT to enhance its antitumor activity (Appendix A). Similarly, HCC09-0913 tumors were significantly smaller when treated with RT/Vinorelbine compared with Vinorelbine treatment alone (*p* = 0.001), suggesting that the combination therapy was also effective against tumors that are resistant to both RT and Vinorelbine (Figure 3B).

Vinorelbine did not induce an observable manifestation of toxicity when combined with RT, as evidenced by the similar mean mouse body weight in the four treatment groups over the course of the study.

Relative to the control, Vinorelbine-treated HCC19-0913 tumors had a 2.1–3.2-fold higher proportion of p-Histone H3 Ser10-positive cells (*p* = 0.00023), suggesting the ability of Vinorelbine to arrest cells in mitosis [26]. Interestingly, RT significantly reduced p-Histone H3 Ser10-positive cells (*p* = 0.0095) and attenuated Vinorelbine-induced upregulation of p-Histone H3 Ser10 (Figure 3C,D). Vinorelbine was more potent than RT in inducing apoptosis, as determined by the percentage of cleaved PARP-positive cells (*p* = 0.0046). In contrast, RT was more potent in inducing DNA damage than Vinorelbine, as indicated by γH2AX staining (Figure 3C,D). The combination of RT/Vinorelbine also caused more DNA damage than RT alone, as indicated by a more intense γH2AX staining (Figure 3C). The RT/Vinorelbine combination showed 1.8–2-fold (*p* = 0.0005) and 3-fold (*p* = 0.00015) more cleaved PARP-positive cells than Vinorelbine and RT treatment alone, respectively (Figure 3D). Similar data were obtained when HCC30-0805B and HCC09-0913 tumors were analyzed (Appendix A). Compared to non-irradiated or Vinorelbine-treated tumors, RT slightly increased the blood vessel density in HCC30-0805B but not in HCC09-0913 tumors (Figure 3C; *p* = 0.024).

RT increased the levels of DNA damage signaling protein (p-ATR, p-ATM, p-Chk2, and γH2AX), but a decrease in the levels of receptor tyrosine kinase (RTK) pathway proteins (FGFR-1, FGFR-2, FGFR-4). In contrast, Vinorelbine caused a reduction in both DNA damage and RTK proteins, but an increase in γH2AX. Compared with RT or Vinorelbine alone, the combination treatment led to a further increase in cleaved caspase 3 and γH2AX, and a further reduction in the levels of survivin and RTK signaling pathway. Reductions in the proteins involved in cell cycle and proliferation (Cdc25C, Aurora B, p-Cdc2) were also observed in the combination treatment (Figure 4).

## 3. Discussion

HCC is a leading cause of cancer mortality worldwide. It is prevalent in the less developed parts of the world, with most patients presenting with unresectable disease. An established strategy to address inoperable cancers, in general, has been to combine RT with systemic agents with the aim to improve tumor control and clinical outcome. In inoperable HCC, radiosensitizers, such as 5-Fluorouracil, Doxorubicin, Thalidomide, and Capecitabine, with RT have been reported with modest survival rates of around a year [27,28,29,30,31,32,33]. Vinorelbine’s radiosensitizer characteristics have mostly been demonstrated in lung cancer, where it confers survival benefit for patients with inoperable disease treated with combination chemotherapy and radiotherapy [21,22,23,24]. This has, however, not been investigated in HCC.

In this study, we investigated the effects of RT and Vinorelbine on HCC PDX lines as monotherapies and in combination. With RT, we were able to classify 3/16 (18.75%), 7/16 (43.75%), and 6/16 (37.5%) of our HCC PDX lines to be sensitive, moderately sensitive, and resistant to 8 Gy RT, respectively. We also demonstrated that RT caused G2/M cell cycle arrest and apoptosis in vitro and that Vinorelbine augmented the efficacy of RT to induce cell death relative to monotherapies. Mechanistically, RT induced G2/M cell cycle arrest and apoptosis through the reduction of p-Cdc2, with a concomitant increase in p27.

The RT/Vinorelbine combination led to increased inhibition of DNA damage repair, resulting in increased levels of cleaved caspase 3 and γH2AX. Fibroblast growth factor receptors (FGFRs) have been shown to play an important role in cell proliferation, apoptosis, differentiation, and metastases. In our study, we demonstrated FGFR downregulation with the RT/Vinorelbine combination. Hence, the enhanced inhibition of DNA damage repair coupled with the inhibition of FGFR expression and a significant increase in apoptosis may contribute to the potent antitumor activity observed in the combination treatment.

We also evaluated the therapeutic potential of RT and Vinorelbine in HCC PDX models with different levels of RT or Vinorelbine sensitivity. Vinorelbine significantly augmented the apoptotic and antitumor activity across all HCC PDX models when compared with RT or Vinorelbine alone. For example, in the RT-sensitive HCC19-0913, nearly sensitive HCC13-0109, and moderately sensitive HCC13-0212 and HCC01-0909 models, the combination inhibited tumor growth more robustly. The combination was also effective against HCC09-0913 tumors that are resistant to RT and Vinorelbine. This was further corroborated by the decrease in cell proliferation and the increase in apoptosis, resulting in potent antitumor efficacy. Importantly, significant toxicity was not observed with this treatment approach, as evidenced by mice body weight and blood parameters.

The increase in intratumoral blood vessel density following irradiation was presumably due to RT-induced accumulation of bone marrow-derived cells (BMDCs) recruited from the adjacent tissues. An accumulation of BMDCs in tumors has been shown to stimulate new vessel formation and recovery of the vasculature [34,35]. Although the pro-vascular effect from recruited BMDCs has been implicated in tumor-protection and disease relapse [35,36], the increased vessel density by RT and the combination of RT/Vinorelbine may be explained by the downregulation of the FGFR signaling pathway, which contributes to the improved tumor response to RT/Vinorelbine. FGFRs are particularly important RTK signaling proteins, with FGFR-2 and FGFR-3 overexpression contributing to tumorigenesis and poor prognosis in advanced HCC [37,38]. Our previous work, which demonstrated the inhibition of the FGFR signaling pathway in HCC xenografts by Infigratinib, an FGFR kinase inhibitor, led to increased tumoral blood vessel density and decreased tumor hypoxia via vascular normalization [39]. Further investigation to better understand the interactions between Vinorelbine and RT on tumor response and the other cellular mechanisms and signaling pathways underlying its antitumor and antiangiogenic activities will be important. Based on our data, we proposed the molecular mechanism of the combination of RT/Vinorelbine (Figure 5), where it increases cell death through the suppression of the DNA repair pathway, FGFR signaling pathway, and inhibition of cell cycle proteins including Cdc25C, Aurora B, p-Cdc2, and survivin.

Taken together, this study, using clinically relevant HCC tumor models and treatment scenarios, provides evidence to indicate that Vinorelbine and RT improve tumor response without additional toxicity. The addition of sub-toxic levels of Vinorelbine to standard RT is particularly relevant in patients with advanced HCC who have low tolerance for treatment due to abnormal liver function. The clinical significance of this study is further amplified by the cost-effectiveness and accessibility of this approach by utilizing two established, inexpensive, and simple-to-deliver treatment modalities that are readily available in most settings globally. This study, therefore, provides a strong rationale for future early-phase clinical trials aimed at improving the efficacy of frontline therapy for patients with unresectable HCC.

## 4. Materials and Methods

### 4.1. In Vitro Irradiation

Primary organoid cultures (Appendix B) were irradiated using the MDS Nordion GC-40 Cesium 137 gamma source irradiator. HCC cells were considered to be sensitive, moderately sensitive, or resistant to RT when the number of cells was 0–35%, 40–75%, or 80–100% relative to the number of non-irradiated control cells, respectively, at 8 days after irradiation. The potential of the anti-microtubule agent Vinorelbine as a radiosensitizer was also investigated using clonogenic cell survival assay as previously described [40]. Briefly, cells were treated with RT in the absence or presence of 0.05 µM Vinorelbine. Vinorelbine was removed 24 h post-RT, and cells were allowed to grow for another 8 days.

### 4.2. Establishment of the HCC PDX Model and In Vivo Irradiation

This study received ethics approval from the SingHealth and National Cancer Centre Singapore. All animals received humane care according to the criteria outlined in the “Guide for the Care and Use of Laboratory Animals” prepared by the National Academy of Sciences and published by the National Institutes of Health (NIH publication 86-23 revised 1985) [41].

Due to phenotypic variability between PDX lines, several models were used. HCC19-0913 is sensitive to both RT and Vinorelbine; HCC17-0212 is RT-sensitive but Vinorelbine-resistant; HCC13-0109 and HCC01-0909 are moderately sensitive to RT and Vinorelbine; and HCC09-0913 is resistant to both RT and Vinorelbine. HCC PDX lines were subcutaneously implanted on both left and right flanks of male C.B-17 SCID mice aged 9–10 weeks and weighing 23–25 g (InVivos Pte Ltd., Singapore, Singapore) as previously described [25].

The right flanks were locally irradiated, while the left flanks acted as an internal control. Mice were randomly assigned to one of the treatment groups containing 5–8 mice each when the tumor reached 150–200 mm^3^. Mice were anesthetized by intraperitoneal (IP) injection of 5 mg/kg Diazepam and 5 mg/kg Ketamine in saline prior to RT and were immobilized in customized jigs with shielding exposing only the tumor for irradiation. Localized irradiation was delivered with a megavoltage clinical-grade linear accelerator with cone-beam computed tomography (CT) imaging at a dose rate of 200–400 MU/min (Varian Clinac IX, Varian Medical Systems, Palo Alto, CA, USA). The irradiation and Vinorelbine doses were optimized to achieve optimal efficacy and minimal toxicity.

For the dose-response experiment, 5 mice per group bearing the HCC19-0913 xenografts were irradiated with 0, 2, 6, and 8 Gy. For the RT/Vinorelbine combination studies, mice were treated as follows: (1) IP injection with 200 µL vehicle (control), (2) a single irradiation dose of 8 Gy, (3) IP injection with 3 mg/kg Vinorelbine, and (4) combined 8 Gy and IP injected Vinorelbine. The vehicle and Vinorelbine were injected every 3.5 days. Tumor measurements were collected as described (Appendix B).

## 5. Conclusions

Most HCC cases are unresectable upon presentation, and current approaches offer only modest survival benefit for these patients. Vinorelbine and conventional radiotherapy are cost-effective, well-established, and are readily available, making their combination an attractive therapeutic option, especially for patients in developing countries where this disease is prevalent. Therefore, this strategy can potentially address an unmet clinical need, warranting further investigation in early-phase clinical trials.

## Figures and Tables

**Figure 1 cancers-12-00872-f001:**
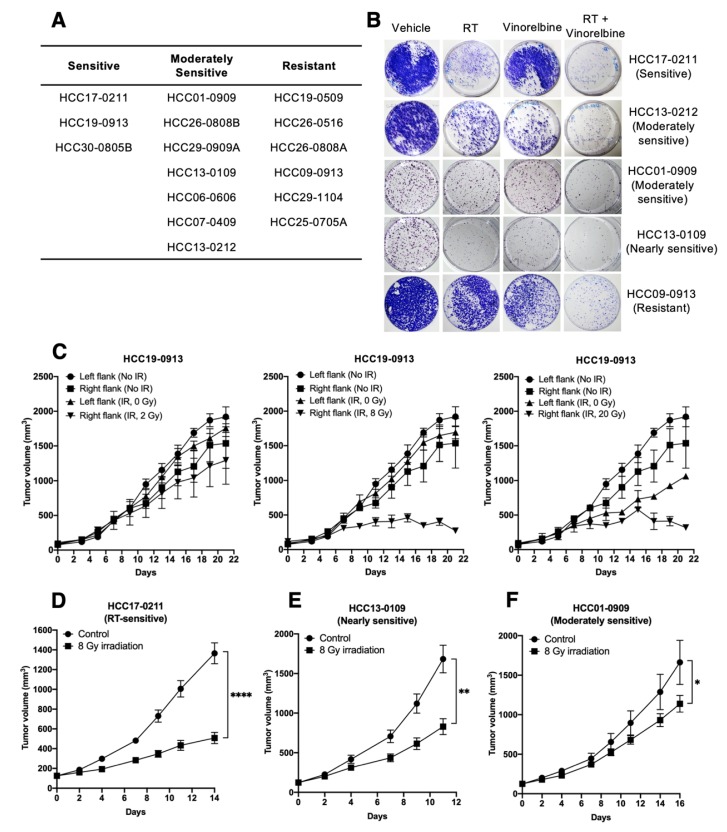
Effects of radiotherapy (RT) on tumor growth in vitro and in vivo. Organoid cultures were prepared as described in Appendix B and subjected to 8 Gy irradiation. Cells were counted 8 days after RT, expressed as percentage of control non-irradiated culture and classified into sensitive, moderately sensitive, or resistant to RT, as described in the Materials and Methods section (**A**). The indicated cultures were subjected to 0 or 6 Gy irradiation in the absence or presence of 0.05 µM Vinorelbine for 24 h. Representative images of colonies are shown (**B**). Mice were subcutaneously implanted with HCC19-0913 tumors on both flanks, with the left flank non-irradiated (internal control) and right flank tumor locally irradiated with a single dose of 2, 8, and 20 Gy (*n* = 6 mice per group); 8 Gy was sufficient to significantly inhibit tumor growth without affecting the internal control and was, therefore, used in subsequent studies (**C**). Mice implanted with the indicated patient-derived xenograft (PDX) lines were treated with 8 Gy RT (**D**–**F**). Tumors were allowed to grow post-irradiation and tumor volumes ± SE plotted (**C**–**F**). Mean tumor weight ± SE at the endpoint are shown (**D**–**F**). * *p* < 0.05; ** *p* < 0.01; **** *p* < 0.0001, Student’s *t*-test. SE, standard error of the mean. IR, irradiation.

**Figure 2 cancers-12-00872-f002:**
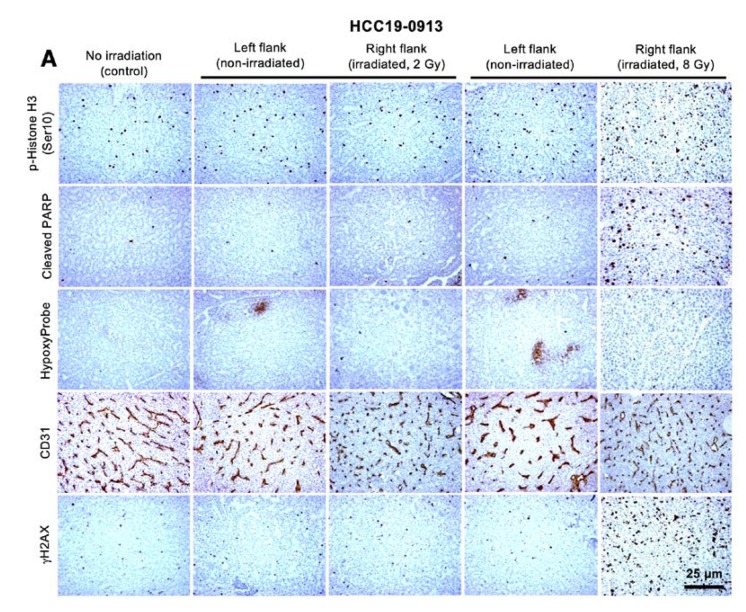
Effects of localized radiotherapy on angiogenesis, cell proliferation, apoptosis, and blood vessels in HCC19-0913 tumors. HCC19-0913 tumors were implanted on both flanks, and tumors on the right flank were irradiated with either 2 or 8 Gy. Tumor tissues were collected 2 days post-irradiation and subjected to immunohistochemistry. Representative images of tumor sections from control non-irradiated mice, internal control (left flanks), and irradiated tumors (right flanks) were stained for blood vessels (CD31), p-Histone H3 Ser 10, cleaved PARP, γH2AX, and Hypoxyprobe as described in Appendix B (**A**). The number of staining-positive cells among at least 500 cells per region was counted and is expressed as the number of positive cells per 1000 cells ± SE (**B**–**D**). For the quantification of mean microvessel density, five random fields at a magnification of ×100 were selected for each section. The number of CD31-positive blood vessels per field was counted as described in Appendix B and presented as mean ± SE (**E**). ** *p* < 0.01; **** *p* < 0.0001, Student’s *t*-test.

**Figure 3 cancers-12-00872-f003:**
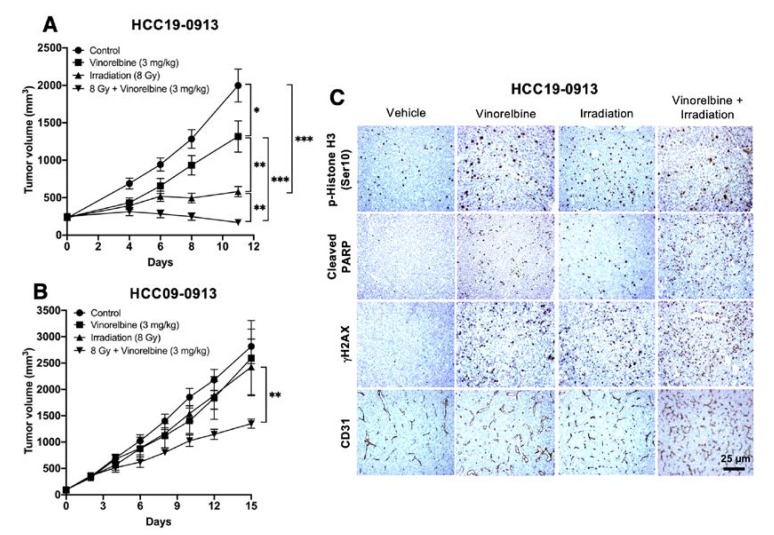
Effects of localized RT, Vinorelbine, and RT/Vinorelbine on tumor growth of the HCC models. HCC19-0913 or HCC09-0913 tumors were subcutaneously implanted on both flanks and treated with vehicle, Vinorelbine, 8 Gy irradiation, or the combination of both as described in the Materials and Methods section. For irradiated mice, tumors on the right flanks were locally irradiated with 8 Gy, and the left flank tumor served as an internal control. Mean tumor volumes ± SE over time are shown (**A**,**B**). Immunohistological analysis of tumors stained with CD31, p-Histone H3 Ser10, cleaved PARP, and γH2AX as described in Appendix B (**C**). Representative images are shown. For p-Histone H3 Ser10, cleaved PARP, and γH2AΧ, the number of staining-positive cells among at least 500 cells per region was counted, as described in the Appendix A and Methods sections, and is expressed as the number of positive cells per 1000 cells ± SE. For quantification of mean microvessel density, five random fields at a magnification of ×100 were selected for each section. The mean number of CD31-positive blood vessels per field was counted and expressed as ± SE (**D**). * *p* < 0.05; ** *p* < 0.01; *** *p* < 0.001; **** *p* < 0.0001, Student’s *t*-test.

**Figure 4 cancers-12-00872-f004:**
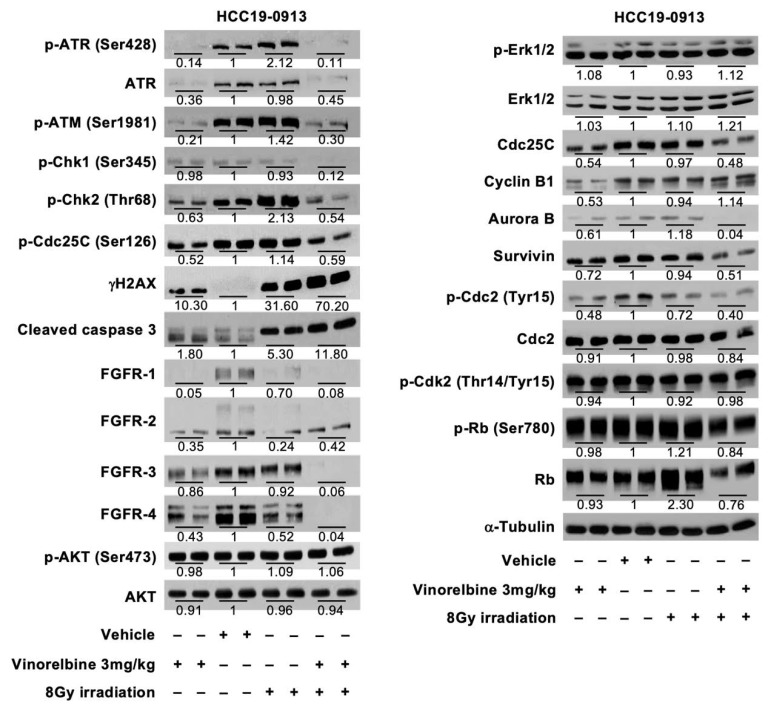
Effects of localized RT, Vinorelbine, and RT/Vinorelbine on the DNA repair pathway, receptor tyrosine kinases (RTKs), and its downstream targets in the HCC19-0913 model. Mice were subcutaneously implanted with HCC19-0913 tumors on both flanks and were irradiated with 8 Gy on the right flank, in the presence or absence of 3 mg/kg Vinorelbine. Tumor tissues were collected on day 2 post-RT, and lysates were subjected to Western blot and quantification analyses, as described in Appendix B. The intensity ratio of each band is expressed as the fold change relative to control. Representative blots are shown.

**Figure 5 cancers-12-00872-f005:**
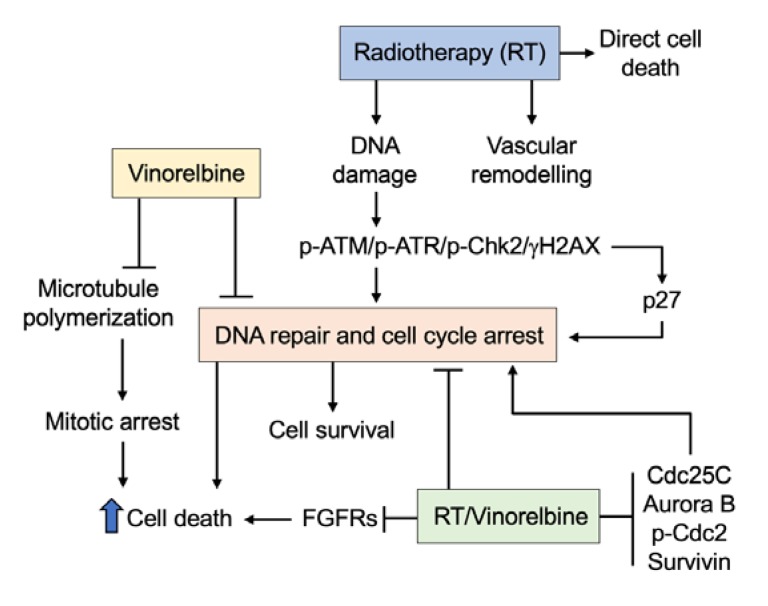
Proposed mechanism for Vinorelbine’s radio sensitizing effect. Radiotherapy induces DNA damage that activates the p-ATM/p-ATR/p-Chk2/γH2AX pathway, which in turn activates the cell cycle control protein, leading to G2/M cell cycle arrest and DNA repair. Under normal circumstances where DNA is repaired, cells continue to proliferate. However, failure to repair damaged DNA leads to cell death. Radiotherapy also induces vascular remodeling by increasing vascular density, which may facilitate the delivery of Vinorelbine. Vinorelbine alone inhibited microtubule polymerization, leading to mitotic arrest and cell death. Furthermore, Vinorelbine inhibited the p-ATM/p-ATR/p-Chk2 pathway, thus reducing cellular DNA repair, which results in increased cell death. Increased levels of DNA damage, the inability to repair DNA, inhibition of FGFR expression, and prolonged mitotic arrest due to inhibition of Cdc25C, Aurora B, survivin, and p-Cdc2 by the combination of RT/Vinorelbine led to more cell death, indicated by elevated levels of cleaved caspase 3.

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
