# Peer review of "Vinorelbine Augments Radiotherapy in Hepatocellular Carcinoma"

_cancers, 2020, doi:10.3390/cancers12040872_

Round 1

Reviewer 1 Report

1. Although Vinorelbine treatment with radiotherapy in hepatocellular carcinoma hasn't been done, the mechanism of the radiosensitization induced by vinorelbine in other human cancer cells has been published. Therefore the current is not initiative. If the authors are to seek for better understanding of the molecular mechanisms on RT and Vinorelbine. The authors should provide a Figure on a proposed mechanism(s) and summary discussion. 

2. The authors should explain why more colonies show up when Vinorelbine was used on HCC17-0211 (Figure 1D)? And at least 2 cell lines per group (sensitive, moderately, and resistant) should be used in Figure 1D.

3. It is unclear how do the primary organoid cultures were prepared.

4. In Figure 2 A-C, what is the difference between no IR (left flank) vs. Irradiated (left flank, 0Gy)? And Why is no difference between Irradiated (right flank, 8Gy) vs. Irradiated (right flank, 20Gy)? Again at least 2 cell lines per group (sensitive, moderately, and resistant) should be used in Figure 2.

5. The same is valid for Figure 3. (at least 2 cell lines per group (sensitive, moderately, and resistant) should be used.

6. When combined with RT and Vinorelbine, achieve higher HCC tumor inactivation than would have been expected from the radiosensitizing synergistic effects or additives effects?

7. The proposed summary mechanism(s) should be included in a Figure.

Author Response

1. Although Vinorelbine treatment with radiotherapy in hepatocellular carcinoma hasn't been done, the mechanism of the radiosensitization induced by vinorelbine in other human cancer cells has been published. Therefore the current is not initiative. If the authors are to seek for better understanding of the molecular mechanisms on RT and Vinorelbine. The authors should provide a Figure on a proposed mechanism(s) and summary discussion.

Our response: Thank you for your comments. We agree with the reviewer that there have been studies on combining Vinorelbine with radiation in other human cancer cells such as inoperable lung cancer. To date, there have been no studies carried out on hepatocellular carcinomas (HCC) with this approach.

As suggested by the reviewer, we provided Figure 5 to summarize the discussion and the possible molecular mechanisms on RT and Vinorelbine. Radiotherapy induces DNA damage that activates the p-ATM/p-ATR/p-Chk2/yH2AX pathway, which in turn activates cell cycle control protein leading to G2/M cell cycle arrest and DNA repair occurs. Under normal circumstances where DNA is repaired, cells will continue to proliferate. However, failure to repair damaged DNA will lead to cell death. Radiotherapy also induces vascular remodelling by increasing vascular density, which may facilitate the delivery of vinorelbine. Vinorelbine alone inhibited microtubule polymerization leading to mitotic arrest and cell death. Furthermore, vinorelbine inhibited the p-ATM/p-ATR/p-Chk2 pathway rendering cells unable to repair DNA leading to more cell death. Increased levels of DNA damage, the inability to repair DNA, and inhibition of FGFR expression and prolonged mitotic arrest due to inhibition of Cdc25C, Aurora B, survivin and p-Cdc2 by RT/Vinorelbine combination led to more cell death, indicated by elevated levels of cleaved caspase 3.

2. The authors should explain why more colonies show up when Vinorelbine was used on HCC17-0211 (Figure 1D)? And at least 2 cell lines per group (sensitive, moderately, and resistant) should be used in Figure 1D.

Our response: HCC17-0211 line was very resistant to Vinorelbine. Figure 1B shows no significant difference in colonies when comparing between vehicle and 0.05μM Vinorelbine-treated plates (p=0.8110).

As suggested by reviewer, we included 2 additional lines in Figure 1B. HCC01-0909 is moderately sensitive while HCC13-0109 is nearly sensitive (between moderate and sensitive) to RT.

3. It is unclear how do the primary organoid cultures were prepared.

Our response: Primary organoid cultures were prepared as described under Appendix A. Briefly, PDX tumors were finely minced and washed 3 times with Modified Eagle’s Medium (MEM), followed by incubation in MEM containing 5% fetal bovine serum (FBS) and 5 mg/ml collagenase (Roche Diagnostics Corporations, Indianapolis, IN, USA) at 37°C for 12h. Tumor fragments were harvested by centrifuging at 200xg for 10 min followed by washing 3 times with serum-free MEM and plated at a density of approximately 8-9x107cells per 100 mm tissue culture dish. Cultures were maintained in MEM supplemented with 10% FBS and 1% penicillin-streptomycin and incubated at 37°C with 5% CO2 for 48h.

4. In Figure 2 A-C, what is the difference between no IR (left flank) vs. Irradiated (left flank, 0Gy)? And Why is no difference between Irradiated (right flank, 8Gy) vs. Irradiated (right flank, 20Gy)? Again at least 2 cell lines per group (sensitive, moderately, and resistant) should be used in Figure 2.

Our response:

No IR (left flank) is tumors in the left flank of control non-irradiated mice.

Irradiated (left flank, 0Gy) is non-irradiated tumors in the left flank of irradiated mice. This serves as an internal control. We were able to locally irradiate the tumors as the mice were irradiated using a linear accelerator instead of a laboratory irradiator, which typically exposes the whole body to ionising radiation.

No significant difference in tumor growth inhibition was observed when comparing between 8Gy- and 20Gy-treated groups (p=0.6938) (page 4). This is presumably due to the saturation in response, where higher dose of irradiation does not provide much additional benefit in terms of tumour killing, but may result in more side effects.

5. The same is valid for Figure 3. (at least 2 cell lines per group (sensitive, moderately, and resistant) should be used.

Our response: As suggested by reviewer, we provided immunohistochemical analysis for 2 additional lines. The results are shown in Figures S3 and S4.  HCC30-0805B is sensitive while HCC09-0913 is resistant to RT.

6. When combined with RT and Vinorelbine, achieve higher HCC tumor inactivation than would have been expected from the radiosensitizing synergistic effects or additives effects?

Our response: To determine if 8Gy plus Vinorelbine was additive or synergistic, a graph of equally effective dose pairs for a single effect level (isobologram) was constructed using the median effect dose for 8 Gy, Vinorelbine, and 8Gy plus Vinorelbine on tumor as described by Tallarida et al. (42). As shown in Figure S5, point A represents the dosage of RT alone (estimated to be approximately 6Gy) to achieved approximately 50% tumor growth inhibition while point B symbolized the required Vinorelbine dosage alone (3mg/kg every 3.5 days) to obtained 50% of tumor growth inhibition. Any two-dose combination that falls on or close to the straight line that link point A to B is defined as additive. Any dose combination that falls to the left of the straight line is defined as superadditive/synergistic. Note that the dose pair such as point X (3Gy plus 1.4mg/kg Vinorelbine) in RT-sensitive HCC19-0913 falls on the left side of the straight line and is thus categorized as synergistic.

7. The proposed summary mechanism(s) should be included in a Figure.

Our response: As suggested by the reviewer we provided Figure 5 to summarize the discussion and the possible molecular mechanisms on RT and Vinorelbine. Radiotherapy induces DNA damage that activates the p-ATM/p-ATR/p-Chk2/yH2AX pathway, which in turn activates cell cycle control protein leading to G2/M cell cycle arrest and DNA repair occurs. Under normal circumstances where DNA is repaired, cells will continue to proliferate. However, failure to repair damaged DNA will lead to cell death. Radiotherapy also induces vascular remodelling by increasing vascular density, which may facilitate the delivery of vinorelbine. Vinorelbine alone inhibited microtubule polymerization leading to mitotic arrest and cell death. In addition, vinorelbine inhibited the p-ATM/p-ATR/p-Chk2 pathway which inhibits cellular DNA repair resulting in increased cell death. Increased levels of DNA damage, the inability to repair DNA, and inhibition of FGFR expression and prolonged mitotic arrest due to inhibition of Cdc25C, Aurora B, survivin and p-Cdc2 by the RT/Vinorelbine combination led to more cell death, indicated by elevated levels of cleaved caspase 3.

Reviewer 2 Report

The article describes a new combination of Vinorelbine and Radiotherapy for treatment of hepatocellular carcinoma in a pre-clinical setting. The use of Vinorelbine is thoroughly investigated and the authors show substantial proof and controls to make certain claims. However, there are some major and minor points which should be addressed before publication.

Major Points:

Lines 73-76: Figure 1A determines an ‘optimal’ dose. It is not clear what is meant by the ‘optimal dose’ (cell killing, survival?) and it is not indicated how this readout is determined. Please elaborate on the findings and indicate de experimental rationale and set-up in the text.

Lines 76-79: The cell cycle analysis of Figure 1C is not that informative after RT if not in combination or Vinorelbine alone. Suggest to move it to supplemental material. Also, why use 2,4,6Gy now while 8Gy was determined as optimal? Please be consistent with the experimental set-up so results of different readouts can be compared.

Lines 80-81: In Figure 1D it is very difficult to see the different colonies since the plates are very full. They show the difference in tumor cell growth after the different treatments, but it is not possible to judge single colonies. Please repeat experiment so single colonies can be assessed or rephrase the corresponding text.

Please explain the rationale of timing and dosing of vinorelbine. Are IC50 and/or timing experiments performed?

Lines 112–118: 150-fold increase of positive yH2AX cells is rather massive and this is not really clear from the representative pictures. The representative images do indicate otherwise since there are positive cells in the controls as well. In addition, the claim of other morphology after irradiation could be true, however in figure 5 this is not observed. I would argue a mistake in microscopy magnification, which is correct in figure 5. Please verify.

Lines 119-124: The suggestion that RT stimulates new vessel formation and recovery of vasculature is misplaced, based on the representative images. In my opinion, the irradiated tumor samples show fragmented vasculature compared to control. This is again observed in figure 5. The measurement of density in the quantification is not well explained and therefore confusing. Also make the scaling of the IHC pictures consistent between figure 3 and 5.

Figure 4: These markers are well characterized for RT. This figure does not add any additional information not known in literature and the results also come back in figure 6. Remove this experiment from the paper.

Figure 5: Quantification of yH2AX is missing, while there are some conclusions based on this result. Moreover, in combination with Figure S2, the magnification of microscopy seems to be rather random, generating confusing pictures. For example: Cleaved PARP stained samples treated with vehicle have much larger nuclei compared to CD31 stained samples. While it was previously claimed that irradiation enlarges nuclei.

The endpoint of the mouse experiments are not defined. Was it on a specific time after treatment or when tumors reached a specific size? Furthermore, order of the treatments is not specified. Please elaborate.

The discussion, especially the first 3 paragraphs will need rewriting. English grammar is poor and confusing. Furthermore, some claims in the discussion are based on questionable quantifications in the results and should be revised or reformulated.

Minor points:

-English grammar needs to be improved.

-Overall: stick to numbers of Gy when mentioning irradiation dose, not two or eight etc

-Overall: measurements of tumor volumes is not always on day zero? Also, graphs are not structured identically, in figure 2 X-axis starts at -4 while later experiments start at 0 or somewhat (not clear how much) before 0.

-Western blots: the abundance of western blots are very hard to compare between experiments or different tumors since the order or protein changes, this could be made more understandable.

-Include the Kaplan-Meier curves for mouse survival.

Author Response

Major Points

Lines 73-76: Figure 1A determines an ‘optimal’ dose. It is not clear what is meant by the ‘optimal dose’ (cell killing, survival?) and it is not indicated how this readout is determined. Please elaborate on the findings and indicate de experimental rationale and set-up in the text.

Our response: Thank you for your comments. As indicated in 4.1. In vitro irradiation under the Materials and Methods section, HCC cells were considered to be sensitive, moderately sensitive and resistant to RT when the number of cells was 0-35%, 40-75% and 80-100% relative to the number of non-irradiated control cells, respectively at 8 days after irradiation (Page 11).

The “effective dose” or “optimal dose” is defined as the dose that kills more than 75% cells at 8 days after irradiation.

Lines 76-79: The cell cycle analysis of Figure 1C is not that informative after RT if not in combination or Vinorelbine alone. Suggest to move it to supplemental material. Also, why use 2,4,6Gy now while 8Gy was determined as optimal? Please be consistent with the experimental set-up so results of different readouts can be compared.

Our response: As suggested by the reviewer, the cell cycle analysis of Figure 1C was not informative after RT if not in combination or Vinorelbine alone. Therefore, we moved it to Figure S1 under Supplementary Materials. The “effective dose” is defined as the dose that killed more than 75% cells at 8 days after irradiation. Since 8Gy killed more than 75% cells at 8 days after irradiation, this was considered as optimal dose.

Lines 80-81: In Figure 1D it is very difficult to see the different colonies since the plates are very full. They show the difference in tumor cell growth after the different treatments, but it is not possible to judge single colonies. Please repeat experiment so single colonies can be assessed or rephrase the corresponding text.

Our response: We agree with the reviewer that the control plate was crowded and was very difficult to see the different colonies. Since HCC17-0211 rapidly spread on the plates, it is difficult to judge single colonies on day 8 after irradiation. However, we believe that there is a distinct visual difference between the treatments vs. control. It is also quite common for colony formation assays to be presented in such fashion as some cells do not readily form individual colonies.

As suggested by the reviewer, we also repeated experiments using 2 additional lines. As shown in Figure 1B, HCC01-0909 is moderately sensitive to RT while HCC13-0109 is nearly sensitive (between moderate and sensitive) to RT.

Please explain the rationale of timing and dosing of vinorelbine. Are IC50 and/or timing experiments performed?

Our response: For the rationale of timing and dosing of vinorelbine, we did a preliminary study where we compared adding vinorelbine before or after irradiation and we found no difference in the colony numbers. The vinorelbine dose used was based on IC50.

Lines 112–118: 150-fold increase of positive yH2AX cells is rather massive and this is not really clear from the representative pictures. The representative images do indicate otherwise since there are positive cells in the controls as well. In addition, the claim of other morphology after irradiation could be true, however in figure 5 this is not observed. I would argue a mistake in microscopy magnification, which is correct in figure 5. Please verify.

Our response: We would like to thank the reviewer for pointing out a mistake in microscopy magnification. All the pictures in Figure 2A were replaced with new pictures with similar magnification. The number of p-histone H2A-X Ser139-positive cells among at least 500 cells per field was counted and expressed as the number of positive cells per 1,000 cells ± SE. The quantification confirmed that there was approximately 150-fold increase of positive yH2AX cells.

Lines 119-124: The suggestion that RT stimulates new vessel formation and recovery of vasculature is misplaced, based on the representative images. In my opinion, the irradiated tumor samples show fragmented vasculature compared to control. This is again observed in figure 5. The measurement of density in the quantification is not well explained and therefore confusing. Also make the scaling of the IHC pictures consistent between figure 3 and 5.

Our response: As shown in Figure 2A, RT stimulates new vessel formation and recovery of vasculature.

We agree with the reviewer that at high magnification (400X), the blood vessels in irradiated tumors were smaller than those in non-irradiated tumors. To avoid any confusion, all the pictures in Figure 2A were placed with new pictures with similar magnification.

For quantification, 5-µm sections were stained with CD31 (1:100) to assess micro-vessel density. Slides were counterstained with hematoxylin (Sigma Diagnostics, Inc., Livonia, MI, USA) for 10 sec at room temperature.  Images were captured on an Olympus BX60 light microscope (Olympus Corporation, Tokyo, Japan). Five random 0.159-mm2 fields at x100 magnification were captured for each tumor. For the quantification of mean microvessel density in sections stained for CD31, 5 random fields at a magnification of x100 were selected for each tumor.  Data were presented as the mean ± standard error. Differences in the mean micro-vessel density were compared. Student’s t-test was used for comparisons between two groups. P<0.05 was considered to indicate a statistically significant difference (See Appendix A).

The scaling of the IHC pictures in figure 2, 3, S2, S3 and S4 are standardized as 25 µM.

Figure 4: These markers are well characterized for RT. This figure does not add any additional information not known in literature and the results also come back in figure 6. Remove this experiment from the paper.

Our response: As suggested by the reviewer, we removed these markers from Figure 4 and placed them in Figure S1.

Figure 5: Quantification of yH2AX is missing, while there are some conclusions based on this result. Moreover, in combination with Figure S2, the magnification of microscopy seems to be rather random, generating confusing pictures. For example: Cleaved PARP stained samples treated with vehicle have much larger nuclei compared to CD31 stained samples. While it was previously claimed that irradiation enlarges nuclei.

Our response: Quantification of yH2AX was included in Figure 2, 3, S2, S3 and S4. To avoid confusion, we replaced all the pictures in Figure 2, 3, S2, S3 and S4 with similar magnification.

The endpoint of the mouse experiments are not defined. Was it on a specific time after treatment or when tumors reached a specific size? Furthermore, order of the treatments is not specified. Please elaborate.

Our response: For RT/Vinorelbine combination studies, mice were treated as follows: 1) IP injection with 200 µl vehicle (control), 2) a single irradiation dose of 8Gy, 3) IP injection with 3 mg/kg Vinorelbine and 4) combined 8Gy and IP injected Vinorelbine. Vehicle and Vinorelbine were injected 12h after irradiation and repeated once every 3.5 days. Treatment commenced when the tumors reached 150-200 mm3 in size. Bi-dimensional measurements were performed once every 2-3 days and tumor volumes were calculated based on the following formula: Tumor volume = [(length) x (width2) x (pi/6)], and plotted as the means ± standard error of the mean for each treatment group vs. time. Experiments were terminated when tumors in the control group reached 1400-2000 mm3. Body and tumor weights were recorded at the time of sacrifice. The tumors were stored at -80ËšC for later biochemical analysis (page 11, Materials and Methods).

The discussion, especially the first 3 paragraphs will need rewriting. English grammar is poor and confusing. Furthermore, some claims in the discussion are based on questionable quantifications in the results and should be revised or reformulated.

Our resnponse: We have made amendments to the discussion section, in particular rewriting the first 3 paragraphs, according to the reviewer’s suggestions.

Minor points:

-English grammar needs to be improved.

Our response: We have made amendments to the manuscript to improve the grammar and have sent the manuscript to MDPI Author Services for further editing and checking.

-Overall: stick to numbers of Gy when mentioning irradiation dose, not two or eight etc

Our response: We have made the amendment accordingly.

-Overall: measurements of tumor volumes is not always on day zero? Also, graphs are not structured identically, in figure 2 X-axis starts at -4 while later experiments start at 0 or somewhat (not clear how much) before 0.

Our response: As suggested by reviewer, all graphs were restructured identically and measurements of tumor volume started on day zero.

-Western blots: the abundance of western blots are very hard to compare between experiments or different tumors since the order or protein changes, this could be made more understandable.

Our response: To compare the changes in protein were quantified as fellows: Total density of the band corresponding to protein blotting with the indicated antibody was quantified using the GS-900 Calibrated Densitometer and Image LabTM Software 6.0.1 (Bio-Rad Laboratories, Inc., Hercules, CA, USA), normalized to both the tubulin loading control and the appropriate phosphorylated/total protein where applicable, and expressed as the fold change.

To make the interpretation of Western blot more understandable, we have also included explanations for the Western blot results in the text, as well as the quantification of the protein changes in the figures (Figures 4, S1, S2).

-Include the Kaplan-Meier curves for mouse survival.

Our response: The survival end point was not investigated in the present study. Mice in all groups were sacrificed when the tumor size in the control group reached 1.4 to 2 cm3

Round 2

Reviewer 1 Report

Based on the current revision, the authors had made significant improvements and effort to address the comments provided by the reviewer. As a result, this peer-reviewed article is now in good quality for publication. Thank you.

Author Response

We would like to thank Reviewer 1 for the valuable comments to improve our manuscript.

Reviewer 2 Report

Thank you for your comments. As indicated in 4.1. In vitro irradiation under the Materials and Methods section, HCC cells were considered to be sensitive, moderately sensitive and resistant to RT when the number of cells was 0-35%, 40-75% and 80-100% relative to the number of non-irradiated control cells, respectively at 8 days after irradiation (Page 11). The “effective dose” or “optimal dose” is defined as the dose that kills more than 75% cells at 8 days after irradiation.

The use of 8Gy is not explained in the results section, which makes it confusing why you use 8 Gy. In addition, it is not clear why you use 6 Gy in Figure 1B. Furthermore, I would rephrase ‘smaller colonies’ in line 84 to ‘less colonies’. It is hard to see the size of the colonies.

The heading of section 2.1 should be altered. The authors describe both in vitro and in vivo results.

We would like to thank the reviewer for pointing out a mistake in microscopy magnification. All the pictures in Figure 2A were replaced with new pictures with similar magnification. The number of p-histone H2A-X Ser139-positive cells among at least 500 cells per field was counted and expressed as the number of positive cells per 1,000 cells ± SE. The quantification confirmed that there was approximately 150-fold increase of positive yH2AX cells.

The pictures are much clearer now. The quantification seems OK. Perhaps the authors could adjust the histogram-graph to visualize that there are NOT 0 yH2AX positive cells in the other conditions. In the histogram it seems to be 0, however on the pictures there are clearly some positive cells.

As shown in Figure 2A, RT stimulates new vessel formation and recovery of vasculature.
Recovery compared to what? Also, the quantification is still very vague. What is determined as ‘density’? The amount of signal or the amount of positive structures?
The figure 2A shows very low vasculature after 8 Gy irradiation compared to no irradiation, while quantification shows significant difference in the other direction. This is very confusing.

Quantification of yH2AX was included in Figure 2, 3, S2, S3 and S4. To avoid confusion, we replaced all the pictures in Figure 2, 3, S2, S3 and S4 with similar magnification.
Quantification in figure 3 makes much more sense and is very different compared to figure 1. Also, now the vasculature is not different. This is again very confusing. It seems that quantification or imaging is not constant.

Overall: perhaps it would be good to only mention the tumors you investigate and give them other names as the codes. The article and figures are hard to read with all these tumor-codes.

Discussion:

Rewriting has helped with understanding the discussion.

The authors still conclude that nuclei are enlarged while no quantification has been given and the authors admitted errors in scaling of images. These observations should be deleted from the manuscript.

Author Response

The use of 8Gy is not explained in the results section, which makes it confusing why you use 8 Gy. In addition, it is not clear why you use 6 Gy in Figure 1B. Furthermore, I would rephrase ‘smaller colonies’ in line 84 to ‘less colonies’. It is hard to see the size of the colonies.

Our response:

6 Gy was shown to give a maximal inhibition in vitro. This was based on our preliminary study (Line 85-86, data not shown)

8 Gy was used as we did not observe a significant difference between 8 Gy and 20 Gy in terms of tumour inhibition. However, with 20 Gy, we observed RT-associated toxicities as evidenced by significant weight loss in mice. Therefore, we decided to use 8 Gy as it provides nearly identical inhibition, but with minimal toxicity. This has been mentioned in the text:

“no significant growth difference was observed when comparing between 8Gy- and 20Gy-treated groups (p=0.6938). 8Gy was deemed efficacious with minimal RT-associated toxicities in vivo and therefore was chosen for subsequent RT/Vinorelbine combination studies” (Line 115-118).

The description for the colonies has been rephrased to ‘less colonies’ (Line 92)

The heading of section 2.1 should be altered. The authors describe both in vitro and in vivo results.

Our response:

The heading has been changed to:

Screening of organoid cultures in vitro and PDX models in mice for sensitivity to irradiation

The pictures are much clearer now. The quantification seems OK. Perhaps the authors could adjust the histogram-graph to visualize that there are NOT 0 yH2AX positive cells in the other conditions. In the histogram it seems to be 0, however on the pictures there are clearly some positive cells.

Our response:

We have adjusted the histogram scale so it captures the lower levels of yH2AX and cleaved PARP more clearly.

As shown in Figure 2A, RT stimulates new vessel formation and recovery of vasculature.

Recovery compared to what? Also, the quantification is still very vague. What is determined as ‘density’?

The amount of signal or the amount of positive structures?
The figure 2A shows very low vasculature after 8 Gy irradiation compared to no irradiation, while quantification shows a significant difference in the other direction. This is very confusing.

Our response:

The recovery of vasculature refers to the structure of blood vessels compared with the non-irradiated left flank. We have replaced the picture with a clearer representative image from the same tissue. The number of CD31-positive blood vessels were counted, regardless of the structure. As a result, even though no-irradiation control appear to have high density, these are mostly formed by hyperdilated blood vessel that occupy more space.

For the quantification of mean microvessel density, 5 random fields at a magnification of ×100 were selected for each section. The number of CD31-positive of blood vessels per field was counted and expressed ± SE.  (Line 143-146)

Quantification in figure 3 makes much more sense and is very different compared to figure 1. Also, now the vasculature is not different. This is again very confusing. It seems that quantification or imaging is not constant.

Our response:

We have replaced the image with a better representative picture (Figure 3C).

Overall: perhaps it would be good to only mention the tumors you investigate and give them other names as the codes. The article and figures are hard to read with all these tumor-codes.

Our response:

We agree that the tumor codes can be a bit hard to follow, as the codes were derived from the date we received the tumour samples and established the PDX models. This system has been used for many years. Unfortunately, we are unable to give them other names as these codes had been used in our previous publications. Therefore, for consistency reason, we would keep the HCC tumor code as mentioned in the manuscript.

Discussion:

Rewriting has helped with understanding the discussion.

The authors still conclude that nuclei are enlarged while no quantification has been given and the authors admitted errors in scaling of images. These observations should be deleted from the manuscript.

Our response:

As suggested by the reviewer, we have deleted (lines 253-256)

“Interestingly, irradiated HCC19-0913 cells exhibited slightly larger cytoplasm and nuclei than control cells (Figure 2), suggesting that irradiation may also damage mitotic spindle apparatus that regulate cell division and chromosomal segregation leading to impairment of cytokinesis”.